# Reducing the Negative Environmental Impact of Consumerization of IT: An Individual-Level Approach

**Ayodhya Wathuge [1] and Darshana Sedera [2],***

1   Faculty of Business, Law and Arts, Southern Cross University, Bilinga, QLD 4225, Australia; dsedera@swin.edu.au
2   Department of Business Technology and Entrepreneurship, School of Business, Law and Entrepreneurship, Swinburne University of Technology, Hawthorn, VIC 3122, Australia
*   Correspondence: darshana.sedera@scu.edu.au

**Abstract:** The internet plays a pivotal role in Industry 4.0, where it provides the underlying infrastructure to support the substantial growth of digital platforms and systems to deliver a wealth of benefits. However, with the unprecedented growth of internet-based applications in recent history, the internet itself is harming the environment. Focusing on individual green motivation and willingness to pay for the green internet, this study explores one's willingness to reduce internet usage. The study employs a survey experiment that involves 376 respondents. The results show that the most effective strategy to reduce internet usage is to incorporate extrinsic strategies and allow individuals to pay a premium for green internet services. Our work contributes to the Industry 4.0 literature by exploring how the negative environmental effects of consumerization of IT can be minimized. The findings of the study are important for technology-based businesses, policy-makers and individuals seeking to reduce the environmental damage of the internet.

**Keywords:** internet use; environmental sustainability; self-determination theory; survey experiment

## 1. Introduction

The advancement of technology during the Fourth Industrial Revolution has led to a significant digital transformation within various industries [1]. This transformation is attributed to the convergence of the internet and new technologies, which has facilitated a shift in the approach to industrial production. According to a report by the Boston Consulting Group, the growth of Industry 4.0 is being driven by several foundational technological innovations [2]. These include the ability for sensors, machines, workpieces, and information technology systems to communicate with one another, and encompass the digital innovation suite consisting of new technologies such as industrial Internet of Things (IoT), artificial intelligence (AI), big data and analytics, additive manufacturing (3D printing) and cloud-based software platforms [3]. These technologies serve as the digital link between machines and individuals in various business processes [4].

The growing trend of consumerization of information technology (IT), brought about by the Fourth Industrial Revolution, has resulted in an increase in the usage of technology products and services in personal and professional settings. The increasing popularity of IoT devices is a testament to this trend. McKinsey and Company [5], citing Gartner Group show that the number of businesses utilizing IoT devices is projected to rise to 43 billion by 2023, representing a 3% increase from 2018. Furthermore, as reported by Forbes, it is estimated that approximately 152,000 IoT devices connect to the internet every minute, as individuals embrace the advancements of Industry 4.0 through the consumerization of IT.

The rise in the adoption of these technologies is accompanied by a corresponding increase in data consumption. By 2025, it is estimated that 73.1 zettabytes (ZB) of data will be generated, which represents a 422% increase from the 2019 output of 17.3 ZB [6]. These

developments demonstrate the profound impact that consumerization of IT is having on technology usage and data production.

While the proliferation of technology devices and the exponential growth of data production brought about by the consumerization of IT offer numerous benefits to businesses, including improved process efficiency, enhanced decision-making capabilities, and increased profits [7,8], they also have negative environmental impacts [9]. As industrial production becomes increasingly digitized, connecting its IT and operational technology with data-generating sensors and transferring this data to the "Edge" or the "Cloud", it requires a significant amount of electrical energy. According to the International Energy Agency (IEA), businesses already account for 42.5% of global electricity consumption in 2014 [10], and the increase in energy demand from industrial consumers will further strain energy networks. This increase in energy consumption leads to a corresponding rise in carbon dioxide ($CO_2$) emissions. The total internet usage in 2019, with an average of six hours of usage per day [11] has been estimated to be equivalent to 1.2 billion years. It is projected that by 2030, the internet will be responsible for 23% of total $CO_2$ emissions [12]. These developments highlight the need for responsible and sustainable approaches to the adoption and usage of technology in the age of consumerization of IT.

The advancements in technology have led to increased efficiencies; however, the exponential growth in internet usage as a result of Industry 4.0 is predicted to surpass such advancements [13]. To mitigate the adverse environmental impact of the widespread utilization of the internet, it is crucial to regulate its usage at an individual level. The significance of this investigation into the impact of internet usage on the environment is driven by two crucial factors: (i) the central role that the internet plays in daily life, making it challenging for individuals to regulate their own usage and (ii) the current pricing structure that offers nearly unlimited access at a capped rate, which fails to incentivize moderation in usage. This study aims to examine the interplay between motivation and willingness to pay a premium for green internet and their influence on the internet usage behavior of individuals.

Past studies have looked at important technical and non-technical steps [14] at the global, country, organization, and individual levels [15] to minimize environmental harm after identifying potential ecological harm. The majority of the previous research has been focused on the organizational level [16]. However, Loock et al. [17] state that studies at the individual level are also important because individual behaviors on a wide scale also have a considerable impact on the quality of the environment. Although individual-level technical steps such as human–computer-interaction designs [18,19] and gamification techniques [20] have been investigated, non-technical self-regulated and non-self-regulated interventions in reducing internet pollution have been not studied [21]. Hence, this study aims to fill that gap by aiming to identify the effect motivation and price interventions have on individuals' willingness to reduce internet usage.

The organization of the paper is as follows. In the subsequent section, a comprehensive review of the existing literature on environmental degradation and the impact of motivation and pricing on modifying human behavior is presented, culminating in the development of a conceptual framework and corresponding hypotheses. The methodology adopted for the study is then thoroughly explained. The results of the study are presented in a clear and concise manner. A thorough examination of the findings is carried out in Section 5. Finally, the conclusions, including both theoretical and practical implications and limitations, are provided.

## 2. Literature Review

### 2.1. Industry 4.0 and Environmental Damage

The Fourth Industrial Revolution, characterized by the rapid transformation of technology, businesses, and societal patterns and processes in the 21st century, is the result of heightened interconnectivity and smart automation [22]. In 2015, Klaus Schwab, founder and executive chairman of the World Economic Forum, popularized the notion of Indus-

try 4.0. He argues that the changes being observed are not simply limited to increased efficiency but represent a significant transition in the nature of industrial capitalism [23]. The ongoing digitization of traditional industrial and manufacturing processes, as well as the integration of sophisticated smart technology, machine-to-machine communication, and the Internet of Things are leading to fundamental shifts in the global production and distribution network [24].

The Fourth Industrial Revolution has resulted in significant advancements in technology and its integration into various industries and societal processes. The rise of the IoT and machine-to-machine (M2M) connections has enabled the consumerization of information technology and has contributed to the growth of the digital economy [25]. Moreover, this becomes more affordable as the prices drop [5,26]. Predictions indicate that M2M connections will account for half of all global connected devices and connections by 2023, with 14.7 billion M2M connections expected by that time [27]. While the internet plays a crucial role in the success of Industry 4.0, its usage also has a significant negative impact on the environment.

The environmental consequences of adopting new technology and behaviors are frequently understood too late, usually when it is difficult to reverse the acquired technologies and behavioral patterns [28,29]. A similar predicament arises if society continues to speed its transition to an unregulated and ecologically unchecked digital world, where the worldwide COVID-19 epidemic hastens the path facilitated by the Fourth Industrial Revolution [28]. The digital behavior patterns that keep developing have negative environmental impacts. Such impacts need to be revealed and treated, to make a successful transition to a low-carbon and green economy. Studies have demonstrated that the development and deployment of internet infrastructure can result in negative impacts on the environment. The three primary ways in which this occurs are through the emission of $CO_2$, wastage of water and land [28]. The power demands of data centers are driven by the need for electricity for both data processing and cooling, further exacerbating the environmental consequences [30].

However, even with the implementation of more energy-efficient technologies, research indicates that the demand for data centers is increasing rapidly, outpacing efforts to minimize their ecological impact. A study by Masanet et al. [31] found that between 2010 and 2018, global installed storage capacity increased 26-fold, and global data center internet protocol traffic increased 11.5-fold. The authors suggest that proactive policy initiatives will be necessary to improve energy efficiency.

In addition, the rapid increase in internet usage can offset any technological advancements aimed at mitigating the environmental impact of internet infrastructure [13]. As such, strategies aimed at managing individual internet usage are becoming increasingly important.

### 2.2. Changing Human Behavior through Motivation and Pricing

There are three possible pathways for changing an individual's attitude towards environmental damage [32]: (1) directly experiencing the phenomenon—e.g., observing a polythene bag blocking a drainage line; (2) persuasive communications to evoke motivations on environmental damage through awareness programs or social marketing; and (3) induced behaviors—for, e.g., offering financial or other incentives to change one's behavior [32]. Since internet-led environmental pollution is not as immediately evident as the example of the polythene bag blocking a drainage line, only pathways (2) and (3) are viable options to evoke a favorable behavioral change. Therein, using the tenants of the self-determination theory (SDT) and the fundamentals of pricing, a conceptual model (see Figure 1) is developed to derive the relevant hypotheses. Figure 1 denotes the abstract conceptual model, while Figure 2 denotes the elaborated model representing the study constructs aligning with the SDT. Please refer to Section 2.3 for explanations related to Figure 2.

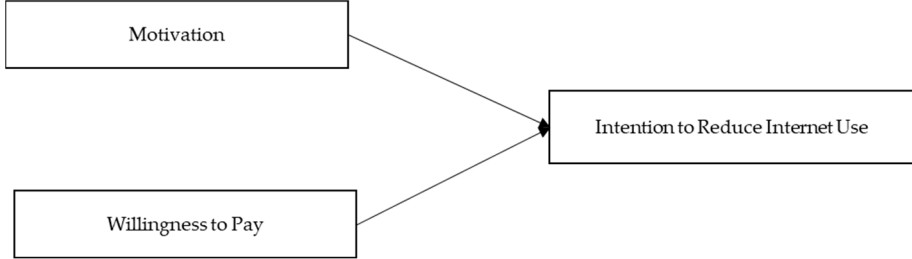

**Figure 1.** Conceptual Model.

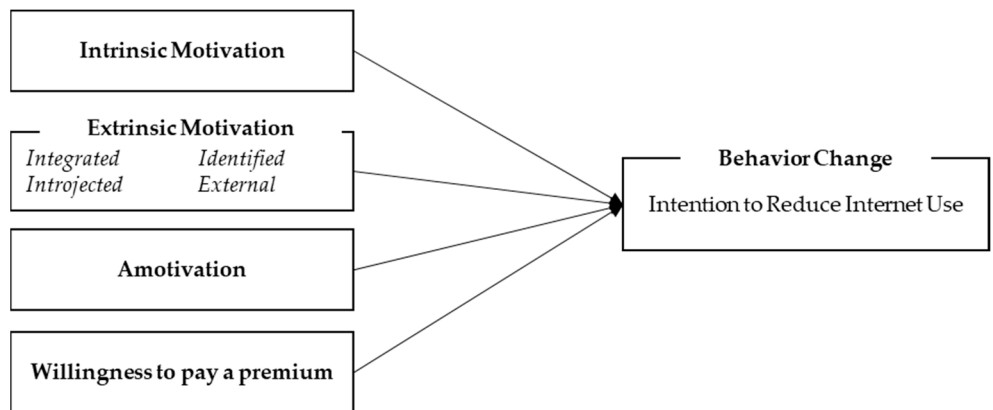

**Figure 2.** The extended model of motivation, willingness to pay a premium and internet use.

### 2.2.1. Motivation to Engage in Pro-Environmental Behavior

Motivation is widely recognized as a powerful tool for shaping human behavior, particularly in the context of green information technology (IT). While previous research has mainly focused on motivation to engage in green IT behavior in an organizational setting, the motivation to engage in green IT behaviors at the individual level has received less attention [33–35]. Despite some studies exploring this topic, most of them have focused on the adoption of transformative green IT services [36–38].

Studies have shown that both intrinsic and extrinsic motivations play significant roles in driving individuals to engage in green IT behavior. However, research has consistently found that intrinsic motivation, which is measured through the pleasure and satisfaction individuals derive from engaging in green IT behavior, is a more significant driver of behavior [36,37,39]. According to the self-determination theory, the highest level of motivation is achieved when an individual experiences pleasure and satisfaction in performing a behavior and feels autonomous, competent, and connected to others. Additionally, while green IT behaviors may also stem from extrinsic motivation [17,40,41], autonomous or intrinsic forms of motivation are believed to play a more prominent role at the individual level [36].

According to the green IT literature, the majority of studies focus on the organization level, where they investigate different types of motivation that affect an organization and its employees in contributing towards reducing damage to the environment [33,34,42]. On the individual level, there are green IT/IS studies that address one's motivation as a viable option for reducing the damage to the environment [36]. However, as per the referred literature, how motivation can be used to reduce environmental damage is not discussed either at the organizational level or at the individual level. While previous research has explored various aspects of green IT behavior, such as the adoption of smart meters and the purchase of eco-friendly IT products [36,37], the intention to engage in green IT behavior by managing one's internet usage has received little attention. As such, this study focuses

on filling this gap, by investigating the role green intrinsic and extrinsic motivations play in reducing individual internet usage.

### 2.2.2. Willingness to Pay for Green Internet

The concept of willingness to pay (WTP) for green products and services has received growing attention in recent years as a means of promoting environmentally sustainable behavior. WTP refers to the amount of money an individual is willing to spend on a product or service that is considered environmentally friendly [43]. In the current context, this refers to one's willingness to pay for environmentally friendly internet data packages.

Numerous studies have investigated WTP for green products and services, with a focus on the factors that influence an individual's WTP, such as their environmental attitudes, knowledge, and personal values [44,45]. Research has found that individuals who are highly concerned about the environment are more likely to be willing to pay a premium for green internet services, compared to those who are less concerned [46]. Additionally, individuals who have a higher level of environmental knowledge are more likely to be willing to pay for green internet services, as they understand the environmental benefits of such services [45]. Personal values also play an important role in determining WTP, with individuals who prioritize environmental conservation being more likely to be willing to pay for green internet services [44].

Several studies have also investigated the relationship between WTP and the perceived environmental performance of green products and services. Research has found that individuals are more likely to be willing to pay for green products and services if they perceive the services to have a high level of environmental performance [47]. Additionally, the perceived social and economic benefits of green internet services have been found to positively influence WTP [43].

Past studies in different contexts have investigated consumers' willingness to pay for green products. Among them, paying a premium to offset carbon in air transportation has attracted much recent attention [48]. The studies show both positive and nonsignificant effects [48]. Apart from that, researchers have investigated the willingness to pay for green packaging [49]. While studies in other contexts have investigated willingness to pay for green products and services, none of the studies referred to in the context of the internet have investigated it. Findings of such a study can be used to inform the development of more sustainable internet infrastructures and to promote the uptake of green internet services by consumers, reducing the negative environmental effects of Industry 4.0. As such, further research is needed to more fully understand the relationship between WTP and green internet services and to inform the development of more sustainable internet infrastructures.

Further to that, it is also important to consider moderators that affect motivation and WTP. In the context of pro-environmental behavior, there exists a divide among studies concerning the influence of demographic factors such as age, gender and income. While some research suggests that these factors have little impact on individual pro-environmental conduct [50], other studies present diverging findings [51]. In addition, investigations have also highlighted the significance of an individual's level of internet usage and the type of internet package utilized (i.e., capped or uncapped data) in determining internet usage patterns [52]. For the current study, we control the effect of demographics on the identified relationships.

### 2.3. Self-Determination Theory

Past studies on motivation to adopt green IT behaviors are mainly based on the technology acceptance model (TAM) and self-determination (SDT) theories. TAM claims that an individual's attitude towards use, which influences behavioral intention, is determined by the salient beliefs of perceived utility and ease of use. The model emphasizes the contrast between extrinsic and intrinsic motivation, with extrinsic motivation defined in terms of its perceived usefulness and intrinsic motivation defined as fun or playfulness [40]. The motivations discussed in TAM focus on quantity. As such, highly motivated individuals

are higher achievers and are more successful than less-motivated ones [53]. On the other hand, SDT discusses the organismic perspective of motivation. It assumes that that people are active organisms with evolved tendencies towards growth, mastery of environmental challenges, and incorporation of new experiences into a coherent sense of self. SDT has a continuum of motivation, where extrinsic motivation is categorized into four further regulations: integration, identified, introjected and external, based on the internalization of the motivation.

As such, SDT allows an in-depth examination of human behavioral change using three types of motivations: intrinsic motivation, extrinsic motivation and amotivation. In relation to the intrinsic and extrinsic motivation, one's behavior can be changed by employing strategies such as providing feedback, motivational interviewing, informational videos, application-based interventions, and feeding behavior information, which have been identified as some fruitful strategies for motivating individuals [54–59]. Such interventions have been used in other contexts as well. An anti-smoking study conducted by Ha and Choi [60] showed that a program based on the degenerative facts related to smoking, such as health-related issues, improved the psychological needs of individuals to reduce smoking, leading to intrinsic motivation. Moreover, anti-smoking apps have been identified as improving intrinsic motivation by presenting intrinsic goal content (health-related information) and extrinsic motivations by presenting extrinsic goal content (money-related information) [61]. In the environmental conservation context, studies have found that educational programs [62], online game-based interventions [63] and social media posts [64] can increase intrinsic motivation.

### 2.4. Hypotheses Development

Internet usage was conceptualized in the control group as the current internet use, as intention to reduce internet usage because of environmental damage cannot be measured without watching the video, due to low awareness of the phenomenon of internet pollution. However, in the experimental group, the internet use construct was conceptualized as the intention to reduce internet use. This was made possible as the subjects in the experimental group were subjected to a treatment that creates awareness of internet pollution. As such, our internet use variable was multi-dimensional. For the analysis purpose, we reverse coded the control group's internet use measurements to a reflect reduction in internet usage. Based on that, following hypotheses were tested in the study.

- Intrinsic Motivation

In intrinsic motivation, a person engages in a behavior due to reasons inherent to the behavior itself, based on values such as pleasure and satisfaction [65]. Herein, positive emotions inspire an individual to engage in pro-environmental behaviors, leading to 'happiness and optimal functioning' [66], which leads to pro-environmental behaviors. However, on the other hand, studies show negative relationships between intrinsic motivation and environmentally friendly behaviors [38,67]. Steg et al. [68] show that people with strong hedonic values (those who seek enjoyment in life) are less likely to engage in pro-environmental behaviors, either because they are not joyful or exciting, or because they impair comfort. As such, we hypothesize that individuals with lower levels of intrinsic motivation to reduce environmental damage may be more likely to consider reducing their internet usage. However, it is undetermined if intrinsic motivation has the same predictive utility in reducing individual internet usage. Hence, we propose:

**Hypothesis 1 (H1).** *When an individual is less intrinsically motivated to reduce the environmental damage, they are more likely to reduce their internet use.*

- Extrinsic Motivation

Extrinsic motivation is when a person engages in a behavior due to reasons external to the activity itself. For example, external rewards like praise, and pressures like social

norms, personal norms and social responsibility manipulate behavior [69]. Ojo et al. [70] examine factors such as social influence and green management culture within an organization, which create external pressure on individuals to engage in green IT behavior. Moreover, studies show that the intention to engage in pro-environmental behaviors increases when extrinsic motivation increases [38,71]. However, some studies show that the self-determination to behave pro-environmentally tends to increase when extrinsic motivation decreases, due to satisfying innate psychological needs [72]. Hence, we propose:

**Hypothesis 2 (H2).** *When an individual is extrinsically motivated to reduce environmental damage, they are more likely to reduce their internet use.*

- Amotivation

Amotivation is the complete absence of motivation. When amotivation increases, engagement in pro-environmental behaviors decreases [73]. According to theory, when interventions that satisfy three psychological needs are employed, an individual moves from being amotivated to being motivated (65). Based on the theory and past studies, this study assumes that there is a connection between amotivation to reduce environmental damage and internet usage. Hence, we propose:

**Hypothesis 3 (H3).** *When individuals are amotivated to reduce their environmental damage, they are more likely to increase their internet use.*

- Willingness to pay

Laroche et al. [44] have noted that there is a lack of clarity regarding the correlation between consumers' willingness to pay a green premium and their engagement in other ecologically favorable behaviors. However, Birgelen et al. [74] attempted to investigate the relationship between pro-environmental behavior (PEB) and consumers' willingness to pay a higher airfare for the purpose of environmental protection, and found a positive association. Moreover, MacKerron et al. [75] conducted a study to examine the willingness to pay for carbon-offset certification among young adults in the United Kingdom. In addition to that, Oreg and Katz-Gerro [76] conducted a research study involving a 27-country sample and found a positive relationship between the willingness to pay and pro-environmental behavior. Considering the evidence from the past literature we hypothesize that individuals who are willing to pay for the green internet are willing to engage in pro-environmental behaviors, that is to reduce internet usage. As such, we hypothesize:

**Hypothesis 4 (H4).** *When willingness to pay increases, the intention to reduce internet use increases.*

- Motivation Crowding Out Effect

Studies also recognize that both intrinsic and extrinsic motivations can, and are likely to, occur together [36]. In particular, studies have observed a lowering of one's intrinsic motivations when controlling interventions such as financial incentives/disincentives are integrated [77]. This is referred to as the 'motivation crowding out' effect [78]. Additionally, studies have also observed crowding out of extrinsic motivation [79]. Motivation crowding out has been reported and tested in different contexts, such as green consumption [80], employee performance [81,82], sharing commercial content in social networking services [83] and environmental conservation [79,84].

They show that when strategies such as premium prices, financial benefits and penalties are introduced, studies have observed a lowering of aspects like self-determination, self-esteem, and voluntariness in performing a task [79,83,84]. In a pro-environmental context, Ref. [79] show that individual premium payments for preserving biodiversity of forests crowd out intrinsic motivation. Moreover, Moros et al. [79] show that extrinsic motivations like guilt and regret were crowded out with the introduction of a payment.

Similarly, we hypothesize that, in the current context when green internet prices are introduced, the existing intrinsic and extrinsic motivations are crowded out, creating more space for pricing. As such, we hypothesize:

**Hypothesis 5 (H5).** *When individuals are willing to pay a premium, intrinsic and extrinsic motivations lose their value.*

- Intention to Reduce Internet Use

Previous research has found positive relationships between environmental motivation and pro-environmental behavior, particularly in green technology adoption or the intention to use green technology [39,40,85]. Accordingly, we hypothesize that environmental motivation correlates positively with the intention to reduce internet usage. Past studies show that intention to engage in sustainable behaviors takes place due to factors such as awareness, motivation and monetary or non-monetary incentives/disincentives. For this study's purpose, we assume that all the other factors except motivation and willingness to pay a premium that affect the intention to reduce internet usage are controlled.

To identify the role of willingness to pay a premium towards green internet services, and intrinsic and extrinsic motivation, we designed the extended study model (see Figure 2) based on the self-determination theory and motivation crowding out theory.

## 3. Materials and Methods

### 3.1. Method

To address the study objectives, a population-based survey–experimental method was selected. This method incorporates characteristics of both survey research and experimental research [86]. It supports generalizability by incorporating a representative sample, and causality by assisting in determining the effect of the treatment. Moreover, this method involves the manipulation of an independent variable, and, due to this, it eliminates the directionality problem [87]. In the population-based survey–experimental method, two pools of subjects are derived, termed the 'control group' and 'experimental group.' Then a 'treatment' is administered to the participants in the experimental group. Therein, this method is similar to a traditional experiment, where observations are made on the variables, upon treatment manipulations.

### 3.2. Study Design

Our control group and the experimental group were 'mutually exclusive' [88]. The purpose of the experiment was to provide a treatment to the experimental group and provide no treatment to the control group. Therein, the experimental group was subjected to an informational video (See details in Appendix A). A tailor-made video of two minutes and 22 s was used as the treatment for the experimental group. The video was created by referring to data from past studies on green-house gas emissions in internet usage. The video was designed to fulfill the three psychological needs of autonomy, competence, and relatedness, needed as the prerequisites for the experimental subjects (see further explanation in Appendix A). Past studies have successfully used videos and environmentally framed behavioral information to satisfy psychological needs [55,57,58,89]. The video included brief descriptions of data centers, popular internet-based activities, their energy requirements, and their effects on the environment. This information was used to improve the autonomy of a subject by increasing rationality and volition of the viewer. The video was pre-tested using two rounds of interviews to ensure that it provides the expected manipulations. The video was subjected to review by a university academic who is an expert in the area, and two PhD students who research environmental sustainability. Further to that, 10 individuals took part in the pilot testing of the video, which resulted in minor changes to the format, graphics, voice, and messaging. The study design is depicted in Figure 3.

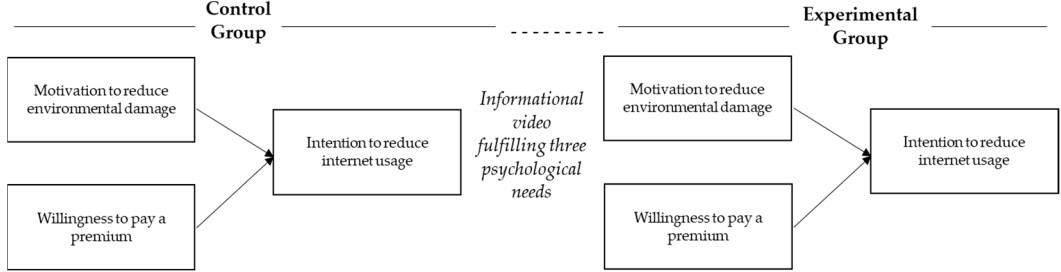

**Figure 3.** Study Design.

All subjects participated in the study voluntarily, and received no compensation. The study was approved by Southern Cross University Human Research Ethics Committee.

### 3.3. Instrument

The survey instruments employed in the study are available in Appendix B. Accordingly, the intention to reduce internet use, willingness to pay a premium and motivation-related variables were measured. The measurement of motivation employed the three sub-constructs discussed above—autonomous, controlled and amotivation [65,73]. Intention to reduce internet usage was investigated using the overall time and data spent on the internet, and measures were derived and adapted from [90]. Measures to capture motivation to reduce environmental damage were derived from the Motivation Toward the Environment Scale (MTES) developed by Pelletier et al. [73], while the measures of willingness to pay a premium were derived and adapted from [44]. The responses were collected using a seven-point Likert scale ranging from 1—Strongly Disagree to 7—Strongly Agree.

### 3.4. Sample

The survey–experimental link was randomly distributed on two social media platforms—Facebook and LinkedIn, with a message to reshare it randomly. The complete random selection of subjects was allocated randomly to the experimental and control groups, using the randomizer tool in the Qualtrics software version 02/2022. A total of 376 usable responses were gathered from 499 responses [91]. Nearly 79% of subjects in the control and experimental groups were between 30 and 56 years of age. According to Statista [92], approximately two thirds of global internet users fall into the given age bracket. As such, the sample approximately represents a major proportion of global internet users, which increases the study's generalizability from the perspective of the age range.

## 4. Results

The structural equation modelling (SEM) approach using partial least squares (PLS) was used for the analysis. The model and constructs followed content and construct validity, confirming the structural model's strength. The PLS-SEM can analyze complex research models, model latent variables, and estimate measurement errors. To evaluate the structural model, the SMART PLS 4.0 software was used with the bootstrap resampling method (4999 resamples). Using regression, we explored the effect of motivation and willingness to pay a green premium with the intention to limit internet use. We evaluated the measurement model first, then tested the structural model, using a two-stage analytical approach.

### 4.1. Measurement Model

Testing for convergent validity and evaluating discriminant validity are both part of the measurement model evaluation process. We examined convergent validity using three indicators: construct composite reliability, average variance extracted (AVE), and item loadings. The constructs' composite reliability (CR) was larger than the required criterion of 0.7. The results indicated follow CR values (values within brackets indicate the control

group): Intrinsic 0.902 (0.888), Integrated 0.897 (0.929), Identified 0.887 (0.897), Introjected 0.885 (0.910), External 0.899 (0.908), Amotivation 0.933 (0.916), Intention to reduce internet use 0.929 (0.910), Willingness to pay 0.896 (not applicable). Moreover, the AVEs of all constructs exceeded the required cut-off of 0.5. The results indicated follow AVE values (values within brackets indicate the control group): Intrinsic 0.756 (0.727), Integrated 0.744 (0.813), Identified 0.724 (0.744), Introjected 0.720 (0.772), External 0.748 (0.767), Amotivation 0.823 (0.787), Intention to reduce internet use 0.724 (0.627), Willingness to Pay 0.742 (not applicable). Apart from that, we also examined the item loadings of each construct, which rendered values higher than the accepted level of 0.7. This was true for both the control group and the experimental group.

Next, we examined the discriminant validity. The degree to which a construct's measurements differ from those of other constructs is referred to as discriminant validity. The Fornell and Larker [93] criterion establishes discriminant validity when the square root of a construct's AVE is greater than its correlation with all other constructs. The square root of AVE for a construct was stronger than its correlation with other constructs in this study. This confirms the discriminant validity for both the groups. Test results are provided in Table 1 for the control group, and in Table 2 for the experimental group.

**Table 1.** Fornell and Larker Test Results—Control group.

| | AM | IU | Ext | Iden | Integ | IM | Intro |
|---|---|---|---|---|---|---|---|
| Amotivation (AM) | 0.887 | | | | | | |
| Intention to reduce internet use (IU) | 0.074 | 0.792 | | | | | |
| External (Ext) | 0.178 | 0.178 | 0.876 | | | | |
| Identified (Iden) | −0.041 | 0.427 | 0.364 | 0.863 | | | |
| Integrated (Integ) | 0.062 | 0.307 | 0.505 | 0.793 | 0.902 | | |
| Intrinsic (IM) | −0.043 | 0.352 | 0.341 | 0.83 | 0.697 | 0.853 | |
| Introjected (Intro) | 0.036 | 0.355 | 0.439 | 0.634 | 0.657 | 0.599 | 0.878 |

**Table 2.** Fornell and Larker Test Results—Experimental group.

| | AM | WIP | Ext | IU | Iden | Integ | IM | Intro |
|---|---|---|---|---|---|---|---|---|
| Amotivation (AM) | 0.907 | | | | | | | |
| WTP | 0.165 | 0.861 | | | | | | |
| External (Ext) | 0.153 | 0.473 | 0.865 | | | | | |
| Intention to reduce internet Use (IU) | 0.218 | 0.416 | 0.465 | 0.851 | | | | |
| Identified (Iden) | 0.013 | 0.382 | 0.292 | 0.326 | 0.851 | | | |
| Integrated (Integ) | 0.036 | 0.523 | 0.397 | 0.311 | 0.612 | 0.862 | | |
| Intrinsic (IM) | −0.069 | 0.403 | 0.283 | 0.171 | 0.707 | 0.666 | 0.869 | |
| Introjected (Intro) | 0.079 | 0.455 | 0.343 | 0.296 | 0.579 | 0.485 | 0.442 | 0.848 |

Validating Higher-Order Constructs (HOC)

The study's higher-order construct was extrinsic motivation, which was based on four lower-order constructs (LOCs): integrated, identified, introjected, and external regulations. Outer weights, outer loadings, and variance inflation factor (VIF) values were used to determine higher-order construct validity. Two outside weights were discovered to be significant [94]. Aside from that, except for the control group's external regulation LOC, outer loadings were determined to be more than 0.5 for each lower-order construct [95]. Hence, all the lower-order constructs were retained for the analysis. Furthermore, VIF values were evaluated, to rule out collinearity. All of the VIF values were lower than the suggested limit of 5 [94]. Based on the above criteria, HOC validity was established. Tables 3 and 4 provide outer weights, outer loadings and VIF values for both the control group and the experimental group.

**Table 3.** Validity of HOC—Control group.

| HOC | LOC | Outer Weight | T Statistic | *p* Values | Outer Loadings | VIF |
|---|---|---|---|---|---|---|
| Extrinsic | Integrated | −0.394 | 1.361 | 0.000 | 0.676 | 3.350 |
| | Identified | 0.991 | 4.324 | 0.000 | 0.947 | 2.905 |
| | Introjected | 0.391 | 1.606 | 0.000 | 0.785 | 1.941 |
| | External | 0.055 | 0.292 | 0.010 | 0.388 | 1.399 |

**Table 4.** Validity of HOC—Experimental group.

| HOC | LOC | Outer Weight | T Statistic | *p* Values | Outer Loadings | VIF |
|---|---|---|---|---|---|---|
| Extrinsic | Integrated | 0.067 | 0.324 | 0.000 | 0.616 | 1.784 |
| | Identified | 0.323 | 1.981 | 0.000 | 0.646 | 1.918 |
| | Introjected | 0.111 | 0.759 | 0.000 | 0.589 | 1.618 |
| | External | 0.752 | 5.794 | 0.000 | 0.911 | 1.231 |

The results indicate that our measurement model is appropriate.

*4.2. Structural Model*

We then examined the structural model, to put our hypotheses to the test. We created three models out of the available data. Table 5 provides a concise description of the nature of the models under consideration, the associated respondent cohort, and the corresponding figure number wherein the pertinent model's statistics are showcased.

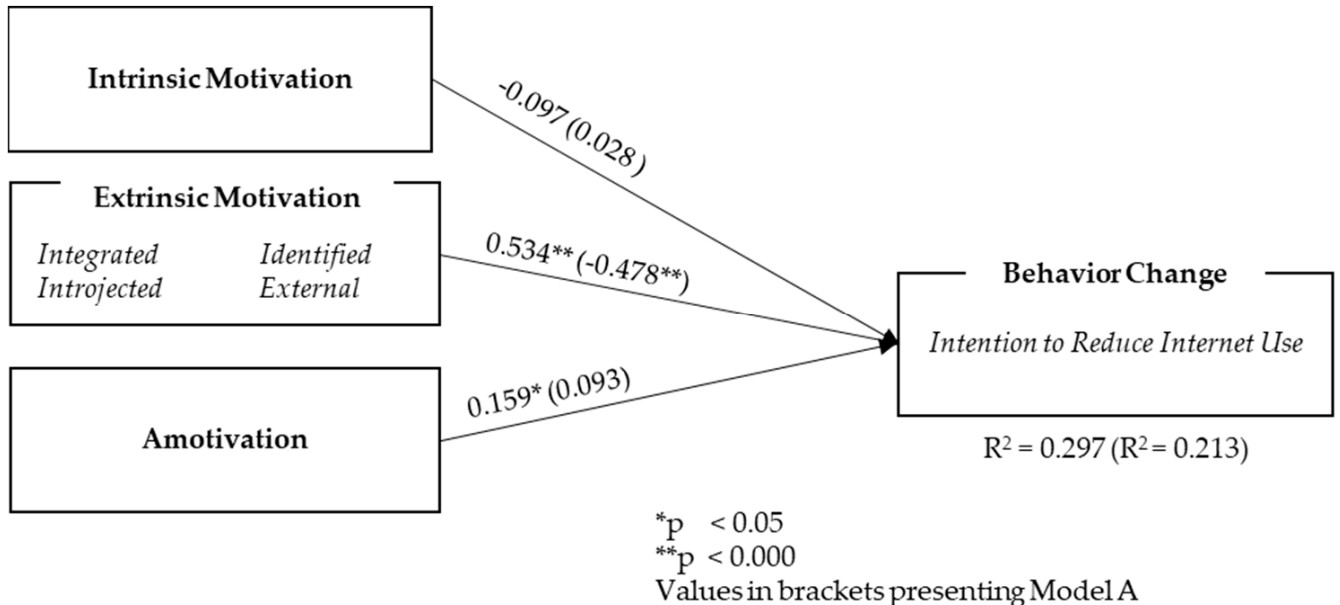

**Figure 4.** Structural Equation Model Depicting the Relationships related to motivation (Models A and B).

In Figure 4 we present the results from our PLS SEM analysis related to motivations. The $R^2$ and path coefficients show the effects of motivations on the intention to reduce internet usage.

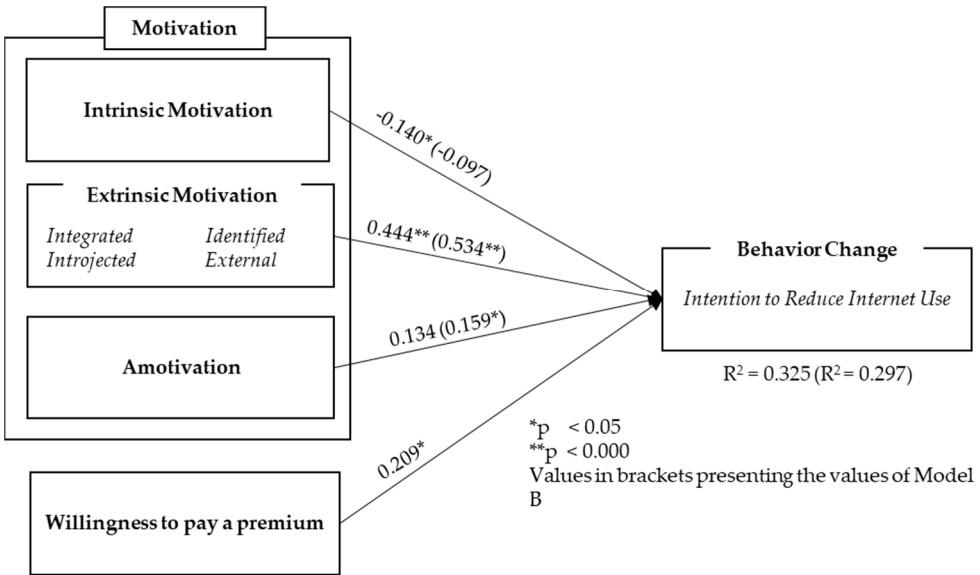

**Figure 5.** Structural Equation Model Depicting Relationships of Models B and C.

**Table 5.** Summary of the Models.

| Model | Description | Group | Figure |
|-------|-------------|-------|--------|
| Model A | Tests the effects of intrinsic and extrinsic motivations and amotivation on intention to reduce internet usage | Control group | Figure 4 |
| Model B | Tests the effects of intrinsic and extrinsic motivations and amotivation on intention to reduce internet usage | Experimental Group | Figures 4 and 5 |
| Model C | Tests the effects of intrinsic, extrinsic motivations, amotivation and willingness to pay on intention to reduce internet usage | Experimental Group | Figure 5 |

The results conclude that, after the informational video, extrinsic motivation still plays the major role in reducing the internet usage of individuals. In the experimental group, extrinsic motivation ($\beta = 0.534$, $p < 0.000$) had a significant positive impact on the dependent variable, thus confirming the H2. Moreover, amotivation ($\beta = 159$, $p < 0.035$) had a significant positive relationship with the intention to reduce internet usage. This result is opposite to what we hypothesized. As such, H3 is not supported for both the control and experimental groups. Furthermore, we could not arrive at a conclusion for the relationship between intrinsic motivation and intention to reduce internet usage for both the study groups. Thus, H1 is not supported. Overall, the individuals' intention to reduce internet usage has increased in the experimental group compared to the control group.

Once the effects of motivations were derived, we added willingness to pay a premium variable, and re-ran the tests. A few significant changes happened in the results. Figure 5 shows the structural equation model depicting the relationships after willingness to pay a premium was added.

Once the pricing variable is added, the results show that individuals are willing to pay a premium ($\beta = 0.209$, $p < 0.011$) while reducing internet usage. Thus, hypothesis 4 was supported. Moreover, significant changes took place in other motivation variables once willingness to pay was added. Specifically, the extrinsic motivation, which played

a major role in the experimental group before adding pricing ($\beta = 0.534$, $p < 0.000$), lost its value ($\beta = 0.444$, $p < 0.000$) with the addition of pricing, thus showing motivation crowding out. On the other hand, intrinsic motivation's negative effect on the intention to reduce internet usage became prominent ($\beta = -0.140$, $p < 0.047$), again showing motivation crowding out. Thus, hypothesis 5 was supported, showing that both intrinsic and extrinsic motivations lost their value after adding the pricing intervention. Furthermore, the effect of amotivation on the intention to reduce internet usage was minimized; however, we were unable to conclude the relationship, as it yielded results as insignificant. When considering the overall effect on the dependent variable, it showed a significant improvement after adding the pricing variable. The explained variance of the intention to reduce internet usage improved to approximately 33% after adding willingness to pay a premium.

## 5. Discussion

Our work contributes to the Industry 4.0 literature by exploring how the negative environmental effects of consumerization of IT can be minimized. Specifically, we discuss how the environmental impact of the backbone of Industry 4.0, the internet, can be minimized by using environmental motivation and willingness to pay a premium. We incorporate self-determination theory and motivation crowding out as theoretical underpinnings of the study, and provide a broad explanation of the results in the following section. Overall results show that the most effective strategy to reduce internet usage is to incorporate extrinsic strategies and allow individuals to pay a premium on green internet service.

The results of the current study provide important insights into the role of motivation in reducing internet usage. The findings suggest that extrinsic motivation plays a major role in reducing internet usage in the experimental group before and after adding pricing. This highlights the importance of external factors, such as personal values systems and internal and external rewards and pressures, in shaping internet usage behaviors. These results contradict the assertions put forth in the literature on green IT, which suggests that intrinsic forms of motivation primarily drive individuals to engage in environmentally friendly behaviors [36,37,96]. Notably, Wunderlich et al. [33] contend that intrinsic motivations played a prominent role in encouraging individuals to utilize a green information system aimed at modifying electricity usage behaviors. This disparity highlights the distinctive nature of internet usage reduction compared to other pro-environmental behaviors, necessitating separate recognition.

Moreover, in the experimental group, before adding pricing, the extrinsic motivation played a prominent role. However, after adding pricing the effect it created diminished, showing the effect of motivation crowding out. This finding has significant implications, as it demonstrates that when individuals are provided with the option to offset the environmental impact of their internet usage feelings of guilt, responsibility, and societal influence that typically drive behavior change are reduced. Consequently, the effectiveness of extrinsic motivations in reducing internet usage is diminished in the presence of a pricing option. This is consistent with the findings of Moros et al. [79], where they show that individual payments crowded out guilt- and regret-based motivations.

Intrinsic motivation showed two important facts. Firstly, the intrinsic motivation and intention to reduce internet usage showed a negative relationship. This suggests that individuals who are intrinsically environmentally motivated are less likely to have the intention to reduce internet usage, even though they are provided with awareness and information to alter their behaviors. This finding contradicts previous research that has shown that intrinsic motivation can be a powerful predictor of behavior change [36]. The results of the present study can be interpreted in the light of the crucial role that the internet plays in the lives of modern individuals, particularly with the advent of Industry 4.0. The internet has become an indispensable tool that individuals utilize on a daily basis. Moreover, some studies have started to refer to the internet as a utility, like electricity, gas and water, depicting its importance [97,98]. Thus, in the current context, the intrinsic motivation representing pleasure in engaging in behavior does not predict the intention to

reduce internet usage. Therein, we can conclude that individuals are naturally unwilling to reduce their internet usage. This further supports treating individual internet usage as a unique context in relation to establishing environmental sustainability.

Secondly, the negative effect of intrinsic motivation became prominent after adding pricing, thus showing motivation crowding out. The results show that once a price factor is added, the lower the intrinsic motivation, the higher the intention to reduce internet usage. That is, when the individuals has an option to pay for their environmental damage, the innate motivation to reduce the damage diminishes. This result is consistent with the findings of past studies on motivation crowding out [79,84]. These findings underscore the distinctive nature of the internet and its integration into individuals' lives, thereby urging researchers to explore novel approaches to reducing environmental damage associated with internet usage, differing from approaches employed in other environmental sustainability contexts.

One of the most interesting findings of the study was the significant positive relationship between willingness to pay a premium for green internet service packages and intention to reduce internet usage, in the experimental group. This shows that, once information is provided on internet pollution, individuals can be motivated to pay for their damage. Moreover, this suggests that individuals who are willing to pay more for environmentally friendly internet services are also more likely to have the intention to reduce their internet usage. This finding has important implications for internet service providers, as it suggests that there is a market for green internet services and that consumers are willing to pay a premium for them. The current study findings align with the previous studies in the environmental sustainability domain, which emphasizes that awareness of consequences improves willingness to pay more for green products and services [99,100]. Further to this, it aligns with green air-travel literature where individuals are willing to pay more for greenhouse gas emission reductions [101].

## 6. Conclusions, Implications and Limitations

The internet's pivotal role in supporting the growth of digital platforms and systems within Industry 4.0 has undeniably brought numerous benefits. However, the exponential expansion of internet-based applications in recent history has also led to adverse environmental consequences. This study has delved into the realm of individual green motivation and the willingness to pay for environmentally friendly internet services, with a specific focus on reducing internet usage. By employing a survey experiment involving 376 respondents, we have uncovered valuable insights. This section serves as a comprehensive summary of the main study findings, providing both theoretical and practical implications, while acknowledging the limitations of our research. Additionally, it identifies future research opportunities that can further advance our understanding of this important subject matter.

### 6.1. Main Findings

In conclusion, the results of the current study suggest that extrinsic motivation plays a major role in reducing internet usage, while willingness to pay a premium for green internet services can also play a significant role. Moreover, the findings highlight the importance of considering the potential for motivation crowding out when designing programs and interventions aimed at reducing internet usage. Overall, the results suggest that a combination of extrinsic motivation with green pricing options would be the most effective approach to reducing internet usage.

### 6.2. Theoretical and Practical Implications

The findings of this study extend the research into Industry 4.0 in several ways. First, our study is the first study to have empirically investigated the concomitant effects of environmental motivation and willingness to pay a premium for green internet services.

As such, it fills an important gap in the Industry 4.0 literature related to strategies that an individual can take to minimize the environmental damage it causes.

Our research revealed evidence on differing perspectives on the significance of motivation in changing one's behavior. Previous research has demonstrated that both intrinsic and extrinsic motives have a considerable impact on behavior, with the majority of them emphasizing the importance of autonomous or intrinsic motivation [36]. However, our findings show marginal evidence to support this view. We identified the fact that only extrinsic motivation creates a positive impact on pro-environmental behavior, while intrinsic motivation showed the extreme opposite effect.

Furthermore, our research found that paying for green services has a substantial and powerful influence [44,102]. The analysis' findings indicated the 'crowding out' effect of both intrinsic and extrinsic motivation when pricing is present. This demonstrates that, when combined with pricing, an individual loses their intrinsic responsibility and extrinsic responsibility attached to personal norms and social norms, in respect to sustainability.

In conclusion, this study provides important insights into the relationship between environmental motivation and willingness to pay a premium for green internet services in the context of Industry 4.0. The findings suggest that extrinsic motivation has a positive impact on pro-environmental behavior, while intrinsic motivation has an extreme opposite effect. Additionally, the results indicate that pricing plays a substantial role in shaping an individual's behavior towards sustainability. These findings have important implications for businesses, policy-makers, and individuals seeking to reduce their environmental impact. By understanding the motivations that drive pro-environmental behavior, it is possible to design more effective strategies to encourage sustainable practices in the digital age.

### 6.3. Limitations and Future Research

This study has some shortcomings that provide possibilities for further investigation. The data for the study were collected during a global pandemic, which may have increased internet use over the world. As a result, future studies must examine the potential negative impact that such extensive use may have on the study findings. Next, the study observed the overall internet use. This could limit the understanding of different types of internet use, such as internet use for work and leisure. The extent to which individuals reduce internet usage in a context of Industry 4.0 could depend on the type of usage. More importantly, the study only observed the intention of individuals to reduce their usage. Although studies have proved that intention to behave leads to exact behavior, some argue for the opposite. As such, this can be tested, to see if individuals truly reduce their internet usage. Further studies can be conducted to explore the effect that motivation and willingness to pay a premium create on the different types of internet uses. Finally, we can suggest investigating panel data related to internet logs and pricing, to objectively study the same phenomenon.

**Author Contributions:** Conceptualization, D.S.; methodology, A.W.; software, D.S. and A.W.; validation, D.S. and A.W.; formal analysis, D.S. and A.W.; investigation, D.S. and A.W.; resources, D.S. and A.W.; data curation, D.S. and A.W.; writing—original draft preparation, A.W.; writing—review and editing, D.S.; visualization, D.S. and A.W.; supervision, D.S.; project administration, D.S. and A.W.; funding acquisition, D.S. and A.W. All authors have read and agreed to the published version of the manuscript.

**Funding:** This research received no external funding.

**Institutional Review Board Statement:** The study was conducted in accordance with the Declaration of Helsinki, and approved by the Institutional Review Board (or Ethics Committee) of Southern Cross University (protocol code: 2021/150 and date of approval: 19/11/2021).

**Informed Consent Statement:** Informed consent was obtained from all subjects involved in the study.

**Data Availability Statement:** Please contact the authors to acquire data.

**Conflicts of Interest:** The authors declare no conflict of interest.

## Appendix A  Video Shown to the Experimental Group

Sample screenshots of the video shown to the experiment group are available below. The video was 2 min and 22 s long and was tailor-made for the experiment and its survey questions. We considered using an existing information video from a third party, but found that such videos lacked the important prerequisite psychological needs of the SDT.

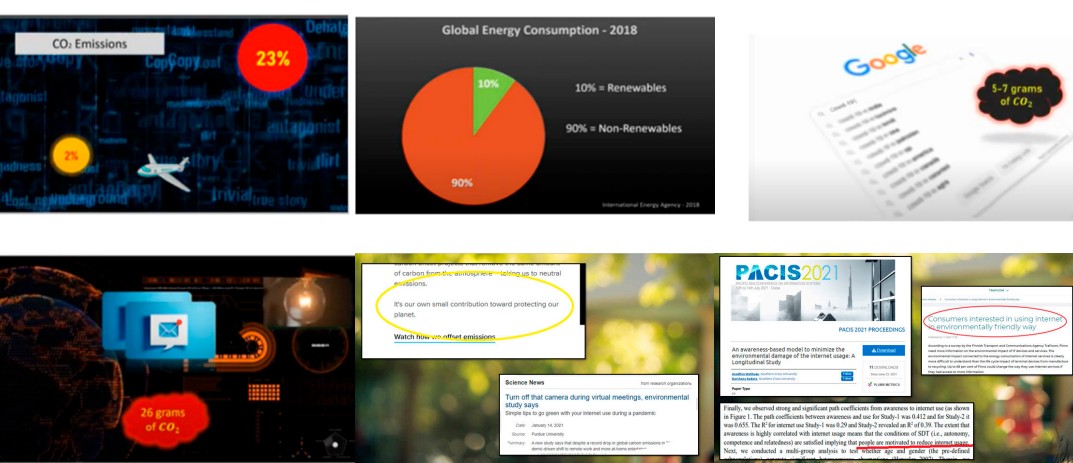

**Figure A1.** Screen Images from the Video.

The three psychological needs are autonomy, competence, and relatedness. The studies show that psychological needs are interdependent; thus, fulfilment or thwarting of one psychological need affects the other. Autonomy is the feeling of control over one's behavior. It creates knowledge about causes and effects and enables people to make better decisions [65]. It was expected that information related to $CO_2$ emissions would provide knowledge of the repercussions of a subject's internet usage and its causes, enabling volitional behavior. By having environmental knowledge, people are likely to perceive the problem as serious and important enough to justify their immediate action [40,103], and this also activates moral obligation and personal norms (leading to internalizing motivation), for performing pro-environmental behavior [104]. As such, people need to possess specific information about the capability of actions taken to start believing that such actions may provide a solution to environmental problems [105]. The knowledge of internet pollution may improve the rationality of individuals and lead consumers to internalizing motivation, because the congruency between behavior and a person's self-identity maximizes the perception of freedom of choice [38]. On the other hand, competence can be defined as a person perceiving himself/herself to be cognitively and affectively related to a particular behavior [65]. As such, the activities an individual performs on the internet and their respective $CO_2$ percentages are expected to make experimental subjects cognitively and affectively related to the actions they perform on the internet. Moreover, steps that a subject can take to reduce the environmental damage of the internet are also provided. As such, the competence in reducing internet usage may increase. Further we included research findings on peoples' likelihood to reduce internet usage to minimize environmental damage. Towards the end of the video, we expect the experiment group's subjects to develop intentions to reduce internet use. Therefore, the display of study findings was expected to increase relatedness, as it shows other people in society have also agreed to reduce internet use to minimize the environmental damage. Krause et al. [106] show that receiving positive feedback from others or using media to reflect on one's own self are mainly associated with benefits for users' self-esteem, which consequently increases the relatedness need. We used the study findings of Wathuge and Sedera [107,108] to satisfy the relatedness need.

The colors, yellow, red and black were used in graphics to signal the severity of the environmental damage of the internet. Moreover, texts were circled and underlined to improve the focus of the subject to specific information.

**Appendix B  Survey Questionnaire**

Internet usage—based on [90]

Experimental Group

I will reduce the time that I spend on the internet every day.

I will reduce the use of the internet.

I will reduce the time that I spend on the internet for leisure activities.

I will reduce the use of the internet for leisure activities.

I will reduce the time that I spend on the internet for work.

I will reduce the use of the internet for work.

Control Group

I spend a lot of time on the internet every day.

I am a heavy user of the internet.

I spend a lot of time on the internet for leisure activities.

I am a heavy user of the internet for leisure activities.

I spend a lot of time on the internet for work.

I am a heavy user of the internet for work.

Willingness to Pay—based on [44] (Experimental group).

If I come across a green internet service, I am willing to switch the internet service, regardless of its price.

I am willing to pay extra for a green internet service than for a non-green internet service.

I am willing to spend extra money per month in order to pay for a green internet service.

Motivation—based on [73] (Experimental and Control group).

I would reduce the potential damage on the environment, because…

Autonomous Motivation

…I take pleasure in conserving the environment.

…I like the feeling when doing something positive for the environment.

…I take pleasure in improving the quality of the environment.

…caring for the environment is an integral part of my life.

…caring for the environment has become a fundamental part of who I am.

…caring for the environment is a part of the way I have chosen to live my life.

…conserving the environment is a sensible thing to do.

…conserving the environment is a reasonable thing to do.

…it is a way I have chosen to contribute to the environment.

Controlled Motivation

…I would regret not doing something for the environment.

…I would feel guilty if I didn't do something for the environment.

…I would feel bad if I didn't do anything for the environment.

…of the recognition I get from others in conserving the environment.

…my family members insist that I do need to conserve the environment.

…my family members will be upset if I don't conserve the environment.

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
