# Peer review of "Reducing the Negative Environmental Impact of Consumerization of IT: An Individual-Level Approach"

_sustainability, doi:10.3390/su151612160_

Round 1
Reviewer 1 Report
Thank you for the opportunity to contribute to improve the paper titled "Reducing the Negative Environmental Impact of Consumerization of IT: An Individual Level Study", with the aim of exploring one’s willingness to reduce internet for environmental sustainability purposes.
Congratulations to the author for their good work.
Abstract: Please consider to reveal some results from your survey.
1. Introduction: Please try to find some academic studies similar / related to yours, so you can present the significance of your study (most current findings, new approach, ...).
Well conducted, updated data presented, relevance of the theme informed, the aim of the study is lacking.
2. Literature Review: Please consider to replace line 161 and put it at the end of line 68. The aim should appear in the Introduction.
Line 267/268 and 270, please use the numeral citation style, not APA one.
Congratulations, the review was very well conducted, with relevant references used, and hypotheses formulated.
3. Materials and Methods
Line 381, please rewrite [78] between brackets.
Line 384, please this fact can contribute to the study's generalization, as other factors were not considered (different countries, economic status, cultural levels...)
4. Results: Well written, with the PLS-SEM well explained.
5. Discussion: Well structured with the contributions well exposed. Limitations informed and future works suggested.
Reference list: Please adjust the reference list to the journal style.
Congratulations, good work.
Author Response
Please see the attached document that outlines how we have addressed reviewer 1 comments.

Reviewer 2 Report
1. The abstract is too confusing in text organization. Please carefully argue below texts: Purpose and research objective, Main findings and results of research (analysis), while authors only reported the background and process?
2. In the Introduction section, what is your motivation? What is your novelty and contributions?
3. From your Fig.2, what is your contribution compared with Fig.1? What gaps the current study can fill? Further more, whether any mediating and moderating variables will affect the mentioned relationships in current hypothesis? If not, why?
4. In 3.2 Study Design, do you think the participants are sufficient for experimental analysis in pilot study?
5. From your Table 1 and 2, please explain why there are many values of correlation coefficient are higher than 0.5? Whether it is right ? Please re-consider it.
6. We suggest that, authors should change the current “5.Discussion” section to “5.Discussion section” and “6.Conclusion, Implications and Limitations section” respectively.
In the “5.Discussion” section, authors should interpret and describe the significance of your findings in light of what was already known about the research problem being investigated, and to explain any new understanding or insights about the problem after you've taken the findings into consideration. The discussion will always connect to the introduction by way of the research questions or hypotheses you posed and the literature you reviewed, but it does not simply repeat or rearrange the introduction; the discussion should always explain how your study has moved the reader's understanding of the research problem forward from where you left them at the end of the introduction.
In the “6. Conclusion, Implications and Limitations” section, authors should start with your main findings in sub-section as "6.1Main findings": author’s original thoughts and evaluation of the obtained results. We suggest authors to add "6.2Theoretical and Practical Implications" sub-section but using brief and concise expressions and logics. Lastly please then state your limitations and future directions as 6.3.
7. Please strictly follow submission guideline of the journal to correct the reference style, more are not in line with the journal request.
None
Author Response
Please see the attached document that outlines how we have addressed reviewer 2 comments.

Reviewer 3 Report
Dear/s Author/s,
Re: Manuscript “Reducing the Negative Environmental Impact of Consumerization of IT: An Individual Level Study”
Reviewer’s report:
The objective of the paper is topical and relevant, as it tries to explore the willingness to reduce the use of the Internet to sustainably protect the environment. The paper is well written, well structured, the experiment is well designed and relevant results are obtained. Only changes are recommended in the abstract, where the four key points are not clearly observed (objective and importance of the topic, methodology, results and conclusions). The hypotheses are not clearly formulated either.
Best regards
Author Response
Please see the attached document that outlines how we have addressed reviewer 3 comments.

Reviewer 4 Report
The article is devoted to the impact of the use of Internet technologies on the environment. Taking up this topic by the authors is justified. The layout of the article and the adopted methodology complies with high academic standards. The discussion on the obtained results is at a high level.
The limitations of the research were presented in a clear and understandable way.
Author Response
Please see the attached document that outlines how we have addressed reviewer 4 comments.

Round 2
Reviewer 2 Report
It can be accepted now if other referees agree.
No
Author Response
All required changes are completed in the revised manuscript. The manuscript also has gone through a professional proofreading exercise.